# Application of Micro- and Nano-Bubbles as a Tool to Improve the Rheological and Microstructural Properties of Formulated Greek-Style Yogurts

**DOI:** 10.3390/foods11040619

**Published:** 2022-02-21

**Authors:** Karthik Sajith Babu, Dylan Zhe Liu, Jayendra K. Amamcharla

**Affiliations:** 1Department of Animal Sciences and Industry, Food Science Institute, Kansas State University, Manhattan, KS 66506, USA; ks7@ksu.edu; 2School of Science, Psychology and Sport, Federation University Australia, Mount Helen, VIC 3350, Australia; dylan.liu@federation.edu.au

**Keywords:** micro- and nano-bubbles, Greek-style yogurt, micellar casein concentrate

## Abstract

The objective of this study was to develop an alternative novel process technology for enhancing the rheological and functional properties of Greek-style yogurt (GSY). The GSY was formulated and prepared in the lab using micellar casein concentrate as a source of protein to achieve a protein content of 10% (*w*/*w*). The changes in physicochemical, microstructural, rheological, and functional properties of control (C-GSY) and micro- and nano-bubbles-treated GSY (MNB-GSY) were studied and compared before and after storage for 1, 2, 3, and 4 weeks. Before storage, the apparent viscosity at 100 s^−1^ (η_100_) was 1.09 Pa·s for C-GSY and 0.71 Pa·s for MNB-GSY. Incorporation of MNBs into GSY significantly (*p* < 0.05) decreased the η_100_ by 30% on 1 week of storage. Additionally, the η_100_ of MNB-GSY was lesser than C-GSY on week 2, 3, and 4 of storage. Notable microstructural changes and significant rheological differences were observed between the C-GSY and MNB-GSY samples. Differences were also noticed in syneresis, which was lower for the MNB-GSY compared with the control. Overall, the incorporation of MNBs into GSY showed considerable improvements in rheological and functional properties. Additionally, it’s a simple, cost-effective process to implement in existing GSY production plants.

## 1. Introduction

Greek-style yogurt (GSY) is traditionally made by straining yogurt to get the necessary total solids content [1]. Presently, for GSY, manufacturing is typically done by concentrating the milk base using ultrafiltration or adding milk protein solids before fermentation, avoiding the creation of acid whey. In recent years, GSY has grown in popularity and market share [2,3]. Protein fortification is one of the common approaches by the dairy industry to make GSY without production of acid whey. The existing process to enhance the protein content of GSY includes the addition of milk protein concentrates (MPC) and micellar casein concentrate (MCC). Due to high protein content, GSY is a nutritionally beneficial dairy product on the market. The flavors of fortified and strained Greek yogurts depend on processing conditions or ingredients. Cooked, burnt/beefy, brothy/potato, dairy sour tastes, and astringency are all common characteristics of fortified Greek yogurts [2]. Additionally, the high protein content in GSY causes an increase in viscosity and subsequently impacts the rheological, functional, and textural properties of the GSYs. MCC and MPC for protein fortification of the yogurt bases are gaining attention in the dairy industry because of their nutritive value, functionality, and sensory attributes [4,5]. Several studies have reported notable changes in yogurts from varying processing conditions [6] and protein addition/fortification [4,7]. Several studies have investigated the effects of fortification of milk with proteins on the physical properties of yogurt [8,9,10,11]. Recent studies have shown several methods available to modify rheological and functional properties of yogurts [12,13,14,15]; however, there is a need for an improved and cost-effective method for enhancing rheological and functional properties of GSY.

Microbubbles (10–50 µm) and nanobubbles (200 nm) are small gas-filled cavities in bulk liquids. In recent years, micro- and nano-bubbles (MNBs) technologies have received substantial attention in industrial applications such as wastewater treatment, cleaning, disinfection, and other agriculture and food-related applications due to the low-cost, eco-friendliness, and the scale-up ease [16,17]. Although the in-depth insights on MNBs and their exact mechanism behind the stabilization are still emerging, their extraordinary longevity has attracted attention in various fields. Increase in the negative charges on the MNB surfaces is due to the increase in hydroxide ion concentration at the gas–liquid interface, thereby reducing the possibility of coalescence of these bubbles and consequently making these bubbles stable in solutions [18]. Generation of MNBs can be accomplished using a bubble-generating agent or by using a bubble generator. The characteristics of MNBs include the increased solubility of gases in liquids, higher zeta potential, greater total surface area-to-volume ratio per mass compared to that of ordinary bubbles [19,20], enabling numerous promising applications. MNBs are negatively charged with an average zeta potential of −30 to −40 mV, depending on the pH of the solution.

The MNBs, while incorporated into food matrices, can improve their physical and textural properties. Zúñiga and Aguilera [21] have reviewed the potentials of introducing MNBs for texture formation/improvements, flavor encapsulation, and delivery of bioactive elements. MNBs were previously used as an alternative for chemical-free cleaning of biofouling in the membrane [22] and stainless steel [23]. MNBs have been widely used for both purifications of water and wastewater [20]. Previous studies have used MNBs for aerobic cultivation in batch fermenters [24]. Moreover, in biomedical application, MNBs are widely used for targeted drug delivery [25] and medical imaging [26]. Zhu et al. [27] reported that nanobubbles (NBs) can prevent the fouling of surfaces and that they can also clean already-fouled surfaces. Very recently, Singh et al. [28] reported CO_2_-MNBs in chlorine and peracetic acid significantly increased the potency of antimicrobial solutions against *Escherichia coli* O157:H7 and *Listeria monocytogenes*. MNBs are capable of reducing the viscosity of concentrated dispersions such as skim milk and milk protein concentrates [29]. Authors also claimed MNBs utilizing a venturi-type hydrodynamic cavitation method displayed impressive results on diverse dairy food systems. Likewise, Phan, Truong, Wang, and Bhandari [30] noted the viscosities of apple juice concentrate, and canola oil was significantly reduced using CO_2_ NBs (size: 50–850 nm). In bulk aqueous solutions, the gaseous MNBs are produced by cavitation, which can be caused by four different mechanisms: hydrodynamic, acoustic, particle, and optical. Very recently, the application of NBs in food processing was reviewed by Phan, Truong, Wang, and Bhandari [31]. In food matrix, which is a complex system, the incorporation of MNBs using a particular cavitation method is still being researched, and their stabilization mechanism is still unclear. The possible advantages of using MNBs in a broad range of applications include the ease with which they can be produced, the low cost of materials, and the potential to easily remove them from the process systems once they have performed their function. Additionally, they may provide solutions to industrial challenges with low environmental impact. By considering the immediate possible commercial applications of MNB technologies in various dairy and other food items, more academic–industrial collaborations can encourage research in this field of science. The objective of this study was to determine the influence of MNB injection on the microstructural, rheological, and functional properties of GSY.

## 2. Materials and Methods

### 2.1. Development of MNB Generation System

Bulk MNBs were generated by a venturi-type bubble generator using filtered atmospheric air. The venturi bubble generator can generate a high number density of MNBs, and the concentration and size of produced bubbles can be regulated by controlling the liquid and air flow rate (0.03 L/min) through the venturi injector (Hydra-Flex, Savage, MN, USA) using the mass flow meter 4140 (TSI, Shoreview, MI, USA). The set-up was equipped with a Ampco AALB-10 positive displacement pump (Glendale, CA, USA) with a flow rate of 2 gallons per min. The generated bubbles in water were investigated using Nanosight in previous research [32]. After MNB treatment, the nanoparticle tracking analysis exhibited a considerable rise in particle concentration, indicating that the hydrodynamic cavitation by venturi injector was effective at producing MNBs. MNB-treated water had ~350 million more bubbles per mL of water tested with a mean size of 249.8 ± 115.8 nm compared to control. Previously, Ahmadi and Khodadadi Darban [33] generated air MNBs using a venturi system (gas flow rate—0.03 L/min) and noted a size of 130 nm. However, no analysis was conducted to measure the bubble size and concentration in the GSY samples. The term MNB was used in the study to account for the microbubbles generated along the NBs, although a clear distinction between NBs and MNBs remains elusive.

### 2.2. Preparation of Formulated Greek-Style Yogurt

The GSY was formulated using TechWizard developed by Owl Software (Lancaster, PA, USA), containing 10% (*w*/*w*) protein and 15% total solids. The base for GSY was made from nonfat dry milk, MCC, and water. Deionized water was heated to 40 °C on a temperature-controlled water bath (Fisher Scientific, Pittsburgh, PA, USA). Subsequently, 7.29% of NFDM and 8.41% MCC were added as the yogurt base to obtain mixtures with a target protein content of 10%. The base was incubated at 40 °C for 30 min with the overhead stirrer (Caframo, Georgian Bluffs, ON, USA) speed set at 500 rpm. It was then kept in a refrigerator at 4 °C for 18 h before use to achieve complete rehydration of MCC powders [34]. Subsequently, on the second day, the base was heated to 90 °C for 10 min and then cooled to 43 °C on a temperature-controlled water bath. The yogurt milk base was inoculated with a commercial yogurt culture, Danisco Yo Mix 495 at 43 °C. The culture was mixed thoroughly and placed in an incubator at 43 °C (Fisher Scientific, Pittsburgh, PA, USA). Fermentation was arrested upon reaching a pH of 4.6 by placing GSY in an ice bath followed by storage in a refrigerator at <4 °C for storage studies.

### 2.3. Experimental Approach

The formulated GSY pumped through the positive displacement pump without attaching the MNB generator was referred to as control GSY (C-GSY) and the formulated GSY passed through the MNB generator was referred to as MNB-treated GSY (MNB-GSY). The C-GSY and MNB-GSY samples were collected and stored <4 °C until further analysis. All the experiments were done in duplicate using independent samples, and each analysis was done in duplicate and the average was used for statistical analysis. The C-GSY and MNB-GSY samples thus produced were evaluated for physical, rheological, microstructural, and functional properties before (week 0), and after storage of 1, 2, 3, and 4 weeks. 

### 2.4. Physicochemical Analysis

Compositional analysis, such as total solids and protein, of the formulated GSY samples were analyzed using gravimetric and Kjeldahl methods, respectively [35]. Titratable acidity, expressed as percentage lactic acid per 100 g of GSY, was performed as per the procedure by Bong and Moraru [4]. The pH of GSY samples were determined using an Accumet benchtop pH meter (Fisherbrand™ Accumet™ AP110, Fisher Scientific, Pittsburgh, PA, USA) at room temperature. The density of the GSY samples was determined by a gravimetric method using Grease pycnometers (Fisher Scientific, Pittsburgh, PA, USA).

### 2.5. Confocal Laser Scanning Microscopy

The microstructure of C-GSY and MNB-GSY were studied using confocal laser scanning microscopy (CLSM), following the method described [36]. The proteins were stained using the Fast green FCF stain (Sigma-Aldrich, St. Louis, MO, USA). Stock solutions of Fast green (5 mg dye in 5 mL water) were applied to the GSY sample for 5–10 min. The stained samples were analyzed in LSM 5 Pa (Zeiss, Thornwood, NY, USA). Three-dimensional images were obtained by scanning the sample across a defined section along the z-axis.

### 2.6. Rheological Measurements

The rheological measurements on the GSYs were performed by rheometer (ATS Rheosystemss, Bordentown, NJ, USA) using a plate and plate geometry (P30CCE) with a 2 mm gap between plates. Flow curves were analyzed at shear rates between 30 and 200 s^−1^. Apparent viscosity (η_100_) was measured at a constant shear rate of 100 s^−1^ for about 100 s. The % lost of structure from initial apparent viscosity (η_0_) and final apparent viscosity (η*_e_*) was calculated using Equation (1). All rheological analyses were performed in duplicate.
(1)$$\%Lost\;of\;structure=\frac{\eta_{0}-\eta_{e}}{\eta_{0}}\times 100$$

### 2.7. Graininess

The graininess of C-GSY and MNB-GSY was analyzed before and during storage time. To capture the digital image, C-GSY and MNB-GSY (1 g) were dispersed in 10 mL of distilled water, then poured into a glass petri dish (Fisher Scientific, Pittsburgh, PA, USA). The samples were pictured by a digital color camera (Sony Corporation, NY, USA) with an optical zoom of 5×. The image analysis was performed and was further analyzed using ImageJ software (Version 1.48; National Institutes of Health, Bethesda, MD, USA). Grains having a diameter >1 mm were computed and expressed as total counts of grains/g of GSY.

### 2.8. Syneresis

Approximately 40 g of samples were transferred to 30 mL centrifuge tubes and were centrifuged at 222× *g* for 10 min, at 4 ± 1 °C. The syneresis was expressed as percent weight relative to the original weight of yogurt.

### 2.9. Water-Holding Capacity

The water-holding capacity (WHC) of C-GSY and MNB-GSY was performed according to the procedure of Singh and Muthukumarappan [37]. For determining the WHC, about 20 g of GSY was centrifuged for 10 min at 669× *g*. The whey expelled (WE) after centrifugation was removed and weighed. The WHC expressed in % was calculated using Equation (2).
(2)$$\mathrm{WHC}\;\left(\%\right)=\frac{\mathrm{GSY}-\mathrm{WE}}{\mathrm{GSY}}\times 100$$

### 2.10. Statistical Analysis

A repeated measures experimental design was used. Changes in the physical, rheological, and functional properties were statistically analyzed using the PROC GLMMIX procedure of SAS (version 9.4, SAS Institute Inc., Cary, NC, USA).

## 3. Results and Discussion

### 3.1. Physicochemical Analysis

The pH of the formulated GSY before storage was 4.50. Formulated GSY had a protein content of 10% (*w*/*w*), and total solids were measured to be 15.13 ± 0.51% (*w*/*w*). The titratable acidity value of the formulated GSY was 1.36 ± 0.01% (expressed as % lactic acid). The titratable acidity of the MNB-GSY (control) was 1.36 ± 0.01% (expressed as % lactic acid). After 4-week storage, it was 1.41 ± 0.01 and 1.45 ± 0.01% (expressed as % lactic acid) for C- and MNB-GSY, respectively. The syneresis rate of the formulated GSY was 11.62 ± 0.39%. The WHC of the formulated GSY was found to be 33.94 ± 1.18%. The density of the MNB-GSY (0.97 g/cm^3^) was lower when compared to C-GSY (1.04 g/cm^3^). The average density of the MNB-GSY was ~6.7% less than C-GSY. Remarkably, Phan, Truong, Wang, and Bhandari [30] reported no significant (*p* > 0.05) changes in densities of NB incorporated liquids (apple juice concentrate and canola oil). The density of the ice cream mix made with NB liquid was 0.77 g/mL, whereas density of the ice cream mix made with regular water was reported be 0.74 g/mL [38]. In a recent study, Adhikari, Truong, Prakash, Bansal, and Bhandari [39] noted that, when compared to the untreated sample, the ice cream treated with 2000 ppm CO_2_ followed by 60 s of acoustic cavitation had considerably (*p* < 0.05) higher overrun values (~88%).

### 3.2. Microstructure

CLSM imaging was used to observe the microstructure of C-GSY and MNB-GSY (before storage). The CLSM 3-D projections of a z-series observed through the microstructure of the C-GSY and MNB-GSY are shown in Figure 1A,B. C-GSY were characterized by having a heterogeneous microstructure, comprising localized dense gel aggregates of protein particles. MNB-GSY protein aggregates appeared to be less dense. Marafon et al. [10] reported that the microstructure of unfortified yogurt showed numerous large pores that were evenly distributed in the protein matrix, whereas the microstructure of yogurt fortified with sodium caseinate at similar protein content showed a more compact protein matrix, with fewer and smaller pores. In the current study, it was noted that the structure of the MNB-GSY is compact compared with C-GSY. Lower-sized MNBs and smaller particles resultant from hydrodynamic cavitation occupy the spaces between the relatively bigger particles, resulting in a lubricating action and a reduction in the GSY viscosity. The bubbles also swell to their maximum size before collapsing forcefully, resulting in a shock wave creation. This generates enough energy to break up the aggregates in the GSY, resulting in a decrease in viscosity and is also evident from the CLSM images. Higher particle size, on the other hand, is linked to higher viscosity [40]. The incorporation of MNBs has the potential to change the physical properties of liquids and increase the mobility of liquid molecules. Indeed, the negatively charged MNBs with an excess of OH-ions can possibly disrupt the unflawed hydrogen-bonding network of the bulk system containing them and can accelerate the mobility of the molecules [18,41]. In a previously published work by Amamcharla, Li, and Liu [29], MNBs have been discovered to lower viscosity of the bulk liquid into which they are injected, particularly liquids containing suspended charged particles.

### 3.3. Rheological Characterization

The rheological properties of C-GSY and MNB-GSY were evaluated at the same protein content (10%, *w*/*w*) because protein content is known to significantly influence the rheological behavior of yogurts [6]. Before storage, the flow curves of MNB-GSY were compared with C-GSY (Figure 2). The ANOVA results showed that the treatment was found to have a significant (*p* < 0.05) effect on η_100_ values. On storage for GSY samples, time and the interaction effect time × treatment were also found to have a significant (*p* < 0.05) effect on η_100_ values (Table 1). The flow curves for the C-GSY and MNB-GSY showed a non-Newtonian shear thinning behavior, with viscosity decreasing as a function of shear rate, characteristic behavior of yogurts. The similar behavior was observed for the GSYs in previous studies by Bong and Moraru [4] and Meletharayil, Metzger, and Patel [12]. Similar flow behavior has been studied and reported for the GSY and plain yogurt in different studies [37,42,43]. At a shear rate of 100 s^−1^, the apparent viscosity was measured and the effect of MNB treatment on the viscosity of GSY before and during storage is shown in Table 2. MNB treatment significantly (*p* < 0.05) reduced the viscosity of GSY samples after passing through the MNB system, decreasing from 1.09 ± 0.08 Pa·s (C-GSY) to 0.71 ± 0.01 Pa·s (MNB-GSY). When compared to C-GSY, this drop in viscosity can be also attributable to the smaller volume occupied by casein aggregates. Previously, Chen et al. [29] noted that the viscosity of the MNB-incorporated GSY showed significant lower viscosity and remained lower throughout the 3-day test period, demonstrating that the MNBs remained stable within the yogurt product. The viscosity lowering effect of NB treatment for a 7-day test period (at 23 and 4 °C) in apple juice concentrate and canola oil was demonstrated by Phan, Truong, Wang, and Bhandari [30], and it was attributed to the NB size and concentration. Körzendörfer, Schäfer, Hinrichs, and Nöbel [15] reported a 40% reduction on η_100_ values of sonicated GSY compared to control GSY. The average η_100_ for the MNB-GSY was ~31% less than C-GSY after week-1 storage. Similarly, Phan, Truong, Wang, and Bhandari [30] noted that the viscosities of apple juice concentrate and canola oil was significantly reduced to 18% and 10%, respectively, after CO_2_ NB treatment (size: 50–850 nm). However, after the week-2, week-3, and week-4 storage, the MNB-GSY samples showed ~58.4%, ~43.1%, and ~50.1%, respectively, lesser η_100_ values compared to the respective weeks C-GSY samples. Likewise, Phan, Truong, Wang, and Bhandari [30] reported the lowering effect on viscosities with NB treatment declines over time as the size of the NBs grew larger and the concentration of dissolved gas decreased. Overall, the MNB treatment had a noticeable effect on the flow behavior of GSY compared with C-GSY. There have been numerous examples of hydrodynamic cavitation being used to reduce viscosity. Previously, hydrodynamic cavitation was observed to be efficient in reducing the viscosity of whey protein concentrate by 20% [44]. The effect of hydrodynamic cavitation on MPCs were investigated for viscosity reduction, and viscosity reductions ranging from 20–56% were observed [45].

The measure of the rate of thixotropic breakdown termed as % lost structure is shown in Table 2. Table 1 suggests that the treatment was found to have no significant (*p* > 0.05) effect on % lost structure, indicative of less severe alterations to the gel structure. However, on storage for GSY samples, the interaction effect time × main effect was found to have a significant (*p* < 0.05) effect on values % lost structure (Table 1). Previously, Besagni, Inzoli, De Guido, and Pellegrini [46] found a link between the bubble size distribution and the viscosity of the liquid phase. After 2 weeks, the % lost structure for the C-GSY and MNB-GSY was 21.01% and 28%, respectively, whereas, after week 4, the % lost structure for the C-GSY and MNB-GSY was 32% and 20.2%, respectively. The viscosity reduction (7–8%) in concentrated protein solutions obtained by hydrodynamic cavitation was found to be continuous over a 2-week storage period [47]. Due to their exceptional lifetime, MNBs have shown to produce stable bulk colloidal suspensions of particle/MNB complexes and prevent aggregation [48,49]. Bulk MNBs do, in fact, play an important role as a buffer between milk protein particles, preventing protein aggregation and interactions over the storage time [29].

### 3.4. Graininess

The grain count of C-GSY and MNB-GSY were studied, and the results are shown in Table 2. Evident visual difference in graininess was observed for the MNB-GSY when compared with C-GSY, before and after storage (Figure 3). The reduction in grain counts observed for MNB-GSY may be attributed to the rearrangements in the protein network. The ANOVA results showed that the treatment was found to have a significant (*p* < 0.05) effect on grain counts. On storage for GSY samples, time and the interaction effect time × treatment were also found to have a significant (*p* < 0.05) effect on grain counts (Table 1). MNB treatment significantly (*p* < 0.05) reduced the grain counts of GSY samples, decreasing from 143.5 ± 11.5 (C-GSY) to 37.5 ± 3.5 grains/g (MNB-GSY). The average grain counts for the MNB-GSY immediately after the treatment was ~73% less than C-GSY. After week-1 storage, the average grain count for the MNB-GSY was ~84% less than C-GSY. Additionally, after the week-2 and week-3, the MNB-GSY samples showed ~87% and ~88%, respectively, significantly lesser (*p* < 0.05) grain counts compared to the respective weeks C-GSY samples. The average grain count of the MNB-GSY was significantly lower (*p* < 0.05), ~98% than C-GSY after week-4 storage. In terms of the influence of storage temperature, the refrigerated condition might slow the growth of NBs. Meletharayil, Metzger, and Patel [12] investigated the effect of hydrodynamic cavitation on GSYs and reported that the commercial GSY had lower (293 grains/g of yogurt), and GSY containing MPC not subjected to cavitation had higher grains (2389 grains/g of yogurt); whereas cavitated GSY (MPC fortified) had lower grains (35 grains/g of yogurt). Remeuf, Mohammed, Sodini, and Tissier [8] observed that fortifying of milk with skim milk powder showed a very low graininess in stirred yogurts (~5 grains/g of sample); whereas, they have also observed that fortifying with WPC had a high level of graininess (~250 grains/g of sample) in stirred yogurts. Additionally, Sodini, Lucas, Tissier, and Corrieu [50] reported that the amount of protein added has a significant effect on graininess. Körzendörfer, Nöbel, and Hinrichs [13] investigated the impact of sonication during yogurt fermentation with starter cultures varying in exopolysaccharide synthesis and reported reduction in graininess. Overall, the MNB injection thus resulted in a steady decrease in the number of grains formed from local stresses developed during the gelation process [51].

### 3.5. Syneresis

Syneresis, known as serum release, is considered as one of the most significant parameters to indicate the quality of yogurt during storage [52]. Table 2 revealed the changes in the syneresis rates of C-GSY and MNB-GSY before and after storage for 1, 2, 3, and 4 weeks. Results in Table 1 suggest that the treatment, time and the interaction effect time × treatment were found to have no significant (*p* > 0.05) effect on syneresis values. This effect may be explained by less structural difference in the gels induced by particle size and rearrangement. It was previously proven that increased syneresis with storage time was usually associated with severe casein network rearrangements [53]. However, before storage, a significant difference (*p* < 0.05) in syneresis was observed between the C-GSY and MNB-GSY (week-0). The average syneresis for the MNB-GSY after MNB treatment was ~37% less than C-GSY. After week-1 storage, the average syneresis for the MNB-GSY was found to be ~47% less than C-GSY. While, after the week-2 and week-3 storage, the MNB-GSY samples showed ~15% and ~28%, lesser syneresis rate as compared to the respective weeks C-GSY samples. The average syneresis for the MNB-GSY was ~22% less than C-GSY after week-4 storage. Indeed, MNB incorporation had improved stability during storage, as syneresis is a function of the stability of the gel matrix. Also, MNB injection possibly helped to establish interpenetrating continuous networks, enhancing the ability of water retaining [54].

### 3.6. Water-Holding Capacity

Table 2 explains the effect of MNB on WHC of stored GSY samples. This property of yogurt to display negligible whey separation is a vital factor for the products retail success and consumer acceptability [6]. The higher WHC was observed in MNB-GSY samples, indicating more water retention ability, whereas lower WHC is associated with excessive rearrangements and unstable gel network [55]. The treatment, time and the interaction effect time × treatment were found to have no significant (*p* > 0.05) effect on WHC. Additionally, no significant difference (*p* > 0.05) in WHC was observed between the C-GSY and MNB-GSY (week-0). The average WHC for the MNB-GSY, after week-1 storage, was ~3.6% less than C-GSY, though it was not significantly different. CLSM images also suggested that reduced WHC is indicative of overall less-cohesive structure and more compact microgel particles [15]. After week-2 storage, the average WHC for the MNB-GSY ~2.9% was higher than C-GSY. The average WHC for the MNB-GSY was ~3.6% higher than C-GSY after week-4 storage. Meletharayil, Metzger, and Patel [12] studied cavitated GSY and reported a significantly higher (*p* < 0.05) WHC compared to control GSYs. The WHC is a clear indicator of the ability of GSYs to retain serum in the gel network. Interestingly, Körzendörfer, Schäfer, Hinrichs, and Nöbel [15] noted that sonication significantly decreased (~4%) the WHC of the stirred yogurt.

## 4. Conclusions

In the present study, we have investigated a method for generating bulk air MNBs using the venturi-type cavitation based on the principle of hydrodynamic cavitation. The custom-designed MNB system could efficiently incorporate MNBs into GSY. Overall, notable changes in functional and rheological behavior of GSY were observed after the MNB injection. The decreased viscosity and graininess of GSY with MNB treatment can be correlated with improved mouthfeel, but in-depth sensory analysis is needed to accomplish conclusive results. The MNB treatment has the potential to be a new processing tool for controlling the viscosity of GSYs, and allows for lower energy use, while having better product functionality. The ability to scale up the process of MNB injection makes this technology promising for large-scale industrial application; although, the exact mechanism behind the stabilization of MNBs in GSY remains unclear. Besides, its eco-friendliness makes the MNB treatment a promising emerging technology. Furthermore, it is expected that this study would be useful for dairy industries in developing a high-efficiency alternative process using MNB injection for modifying and improving the rheological, microstructural, and functional properties of GSYs. 

## Figures and Tables

**Figure 1 foods-11-00619-f001:**
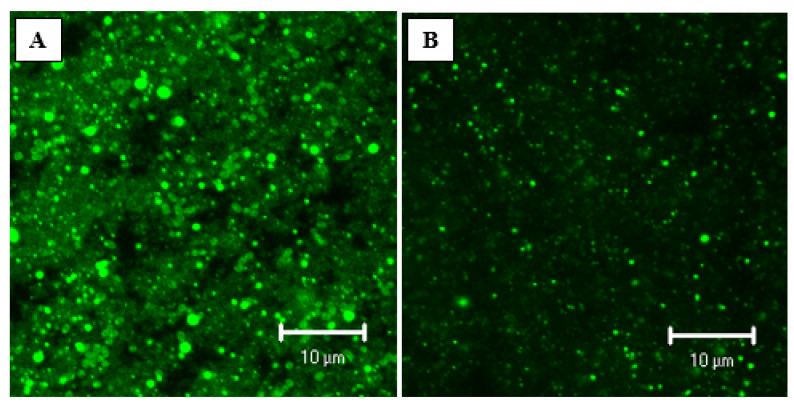
Confocal laser scanning electron microscopy micrographs of Greek-style yogurts (GSY) showing the 3-D projection of a Z-series through the microstructure: (**A**) Control; (**B**) Micro- and nano-bubbles treated GSY.

**Figure 2 foods-11-00619-f002:**
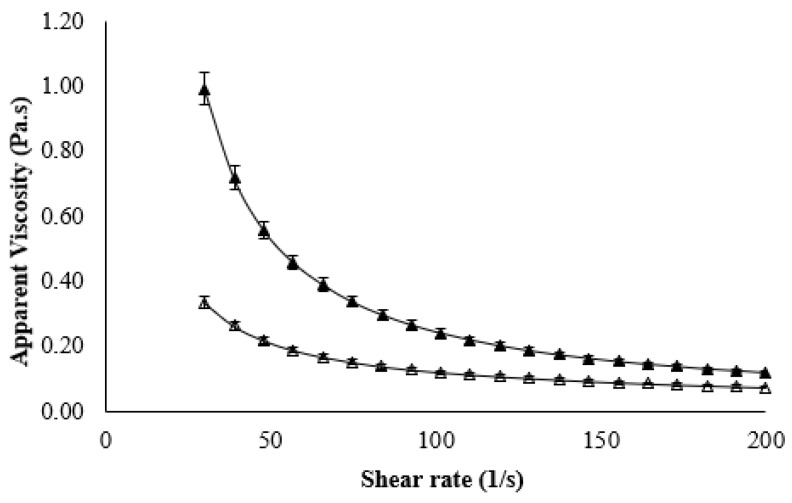
Viscosity as a function of shear rate (flow curves) for Greek-style yogurt (GSY):Control (▲) and Micro- and nano-bubbles (Δ) treated GSY.

**Figure 3 foods-11-00619-f003:**
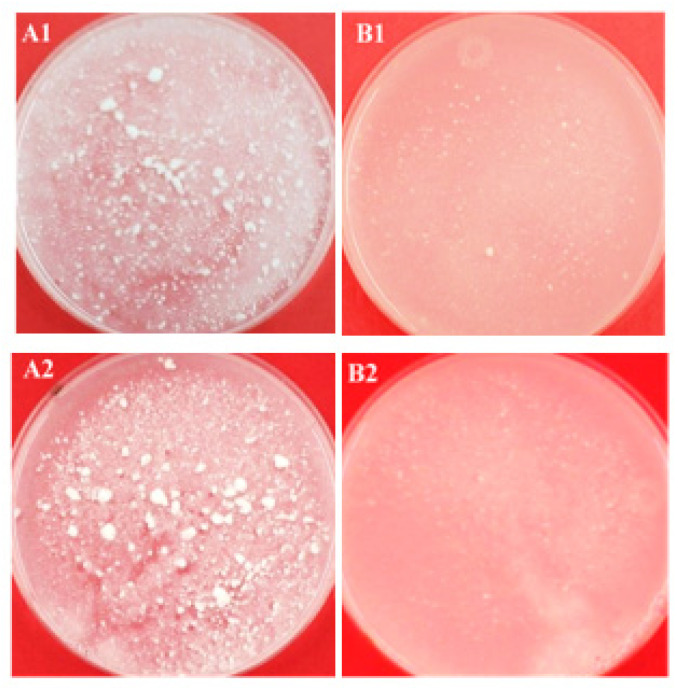
Graininess of Greek-style yogurts (GSY): (**A1**) control GSY before storage; (**A2**) control GSY after 4-weeks of storage; (**B1**) micro- and nano-bubbles (MNB) treated GSY before storage; (**B2**) MNB treated GSY after 4-weeks of storage.

**Table 1 foods-11-00619-t001:** Mean squares (MS) and probabilities (P) of changes in rheological and functional properties in Greek-style yogurt during storage.

Source of Variation	Apparent Viscosity		% Lost Structure		Grain Counts		% Syneresis		% Water Holding Capacity	
	df	MS	df	MS	df	MS	DF	MS	DF	MS
Whole-plot										
Treatment	1	0.91 * (0.003)	1	554.93 (0.06)	1	122,931 * (0.004)	1	50.81 (0.196)	1	0.933 (0.67)
Error	2	0.003	2	42.87	2	490	2	13.9	2	3.9
Sub-plot										
Time	4	0.02 * (0.04)	4	17.02 (0.67)	4	1644 * (0.02)	4	56.6 * (0.0001)	4	5.9 (0.1007)
Time × Treatment	4	0.01(0.16)	4	126.24 * (0.03)	4	1420 * (0.03)	4	0.91 (0.618)	4	2.1 (0.4650)
Error (time)	8	0.006	8	28.57	8	303.1	8	1.31	8	2.12

* Statistically significant (*p* < 0.05).

**Table 2 foods-11-00619-t002:** Characterization of Greek-style yogurts containing micro- and nano-bubbles.

Sample Type	Storage Time	Apparent Viscosity (Pa·s)	% Lost Structure	Grain Counts	Syneresis (%)	Water Holding Capacity (%)
Control Greek-style yogurt	Week-0	1.09 ± 0.08 ^a^	34.14 ± 2.28 ^a^	143.5 ± 11.5 ^a^	8.66 ± 1.74 ^a^	33.18 ± 1.18 ^a^
	Week-1	0.88 ± 0.06 ^a^	38.55 ± 3.22 ^a^	243 ± 11 ^a^	7.36 ± 1.34 ^a^	29.95 ± 0.55 ^a^
	Week-2	1.01 ± 0.11 ^a^	21.01 ± 5.45 ^a^	180 ± 19 ^a^	14.52 ± 0.09 ^a^	29.3 ± 1.01 ^a^
	Week-3	0.98 ± 0.07 ^a^	36.74 ± 7.93 ^a^	182 ± 10 ^a^	16.41 ± 0.34 ^a^	30.95 ± 1.05 ^a^
	Week-4	0.94 ± 0.07 ^a^	32.02 ± 0.14 ^a^	178.5 ± 26.5 ^a^	10.45 ± 1.05 ^a^	31.55 ± 1.85 ^a^
Micro- and nano-bubble-treated Greek-style yogurt	Week-0	0.71 ± 0.01 ^b^	26.11 ± 3.77 ^a^	37.5 ± 3.5 ^b^	5.42 ± 0.54 ^a^	30.77 ± 1.07 ^a^
	Week-1	0.61 ± 0.09 ^a^	17.24 ± 0.32 ^b^	36.5 ± 11.5 ^b^	3.88 ± 0.61 ^a^	28.81 ± 0.31 ^a^
	Week-2	0.42 ± 0.08 ^b^	28.21 ± 5.12 ^a^	23 ± 4 ^b^	12.24 ± 1.09 ^a^	30.15 ± 0.25 ^a^
	Week-3	0.56 ± 0.05 ^b^	18.05 ± 2.53 ^a^	21 ± 5 ^b^	11.81 ± 2.47 ^a^	30.31 ± 1.41 ^a^
	Week-4	0.46 ± 0.01 ^b^	20.18 ± 1.36 ^b^	3.5 ± 0.5 ^b^	8.12 ± 2.28 ^b^	32.71 ± 2.11 ^a^

^a,b^ Mean within a column with different superscript differ (*p* < 0.05) for each storage week; *n* = 4.

## Data Availability

The datasets generated for this study are available on request to the corresponding author.

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
