# Peer review of "Application of Micro- and Nano-Bubbles as a Tool to Improve the Rheological and Microstructural Properties of Formulated Greek-Style Yogurts"

_foods, 2022, doi:10.3390/foods11040619_

Round 1
Reviewer 1 Report
The work relates to an actual technological trend associated with the growing use of micro- and nanobubbles (MNBs) for processing a variety of water-based materials. The task of improving the quality of yoghurts by means of the MNB generation is very practical, since the venturi-type generator used by the authors is relatively cheap. I should note that a thorough comparison of the physical properties of control and MNB-treated samples was carried out. Significant changes resulting from the MNB processing, for instance, decreased graininess and apparent viscosity are explicitly manifested in the tables and graphs, which undoubtedly substantiates the effectiveness of MNB technology in food production.
Below are my comments on the text:
1. In Materials and Methods: With what gas was the yogurt saturated in the MNB generator? Was it CO2? Please specify. What prompted the choice of gas type?
2. Line 11: What does the abbreviation “TS” stand for?
3. Line 122, “reaching a pH of 4.6”: As far as I know, the isoelectric point of caseins belongs to 4.0-5.0. At higher pH, casein has a negative charge. Therefore, at pH = 4.6, casein aggregation is presumably increased. What pH value had the yogurt just before MNB processing? Did the pH of the yogurt change after MNB treatment?
4. Line 243, “new aggregates”: Did you mean to say that new aggregates were formed after MNB processing? Was it casein aggregates or casein/MNB complexes? Proteins show generally good adhesion to bubbles and, thereby, can probably form something like nanofoam in the presence of nanobubbles.
5. Line 252, “after CO2 NB treatment (size: 50-850 nm)”: Here you provide literature data on the size of nanobubbles. Was the size range of nanobubbles in your case the same or different when using your venturi-type generator? In general, the size and concentration of nanobubbles is affected by the type of extrinsic ions dissolved in water and pH, which can be inferred from, for example, the works: Z. Pourkarimi, B. Rezai, M. Noaparast, “Effective parameters on generation of nanobubbles by cavitation method for froth flotation applications,” Physicochem. Probl. Miner. Process. 53(2), 2017, 920−942; S.O. Yurchenko, A.V. Shkirin, B.W. Ninham, A.A. Sychev, V.A. Babenko, N.V. Penkov, N.P. Kryuchkov, N.F. Bunkin, Ion-Specific and Thermal “Effects in the Stabilization of the Gas Nanobubble Phase in Bulk Aqueous Electrolyte Solutions”, Langmuir, 2016, Vol. 32, pp. 11245-11255.
Author Response
Reviewer 1
Comments and Suggestions for Authors
The work relates to an actual technological trend associated with the growing use of micro- and nanobubbles (MNBs) for processing a variety of water-based materials. The task of improving the quality of yoghurts by means of the MNB generation is very practical, since the venturi-type generator used by the authors is relatively cheap. I should note that a thorough comparison of the physical properties of control and MNB-treated samples was carried out. Significant changes resulting from the MNB processing, for instance, decreased graininess and apparent viscosity are explicitly manifested in the tables and graphs, which undoubtedly substantiates the effectiveness of MNB technology in food production.
Below are my comments on the text:
1. In Materials and Methods: With what gas was the yogurt saturated in the MNB generator? Was it CO2? Please specify. What prompted the choice of gas type?
AU: In the current study, we used Air MNBs generated by venturi type bubble generator. Literature suggests use of various gases like CO2, N2, etc. depending on sample and specific intended applications. Using atm air MNBs was the cheapest option. Use of CO2 would change the pH and wasn’t considered as a choice of gas; however, we expect any inert gas would result in similar effects (Line 95-96)
Line 11: What does the abbreviation “TS” stand for?
AU: TS was removed and replaced with total solids throughout the manuscript.
Line 122, “reaching a pH of 4.6”: As far as I know, the isoelectric point of caseins belongs to 4.0-5.0. At higher pH, casein has a negative charge. Therefore, at pH = 4.6, casein aggregation is presumably increased. What pH value had the yogurt just before MNB processing? Did the pH of the yogurt change after MNB treatment?
AU: We stopped the fermentation at 4.6 pH by cooling because the isoelectric point of casein is 4.6 (Körzendörfer, A., Schäfer, J., Hinrichs, J., & Nöbel, S. (2019). Power ultrasound as a tool to improve the processability of protein-enriched fermented milk gels for Greek yogurt manufacture. Journal of dairy science, 102(9), 7826-7837) followed by the MNB treatment. The incorporation of MNB did not significantly impact the titratable acidity (TA) of GSY. The TA of control and MNB treated GSY was included in the manuscript (Line 188).
Line 243, “new aggregates”: Did you mean to say that new aggregates were formed after MNB processing? Was it casein aggregates or casein/MNB complexes? Proteins show generally good adhesion to bubbles and, thereby, can probably form something like nanofoam in the presence of nanobubbles.
AU: Thanks for pointing this out. We modified this as: “smaller volume occupied by casein aggregates”. (Line 251).
Line 252, “after CO2 NB treatment (size: 50-850 nm)”: Here you provide literature data on the size of nanobubbles. Was the size range of nanobubbles in your case the same or different when using your venturi-type generator? In general, the size and concentration of nanobubbles is affected by the type of extrinsic ions dissolved in water and pH, which can be inferred from, for example, the works: Z. Pourkarimi, B. Rezai, M. Noaparast, “Effective parameters on generation of nanobubbles by cavitation method for froth flotation applications,” Physicochem. Probl. Miner. Process. 53(2), 2017, 920−942; S.O. Yurchenko, A.V. Shkirin, B.W. Ninham, A.A. Sychev, V.A. Babenko, N.V. Penkov, N.P. Kryuchkov, N.F. Bunkin, Ion-Specific and Thermal “Effects in the Stabilization of the Gas Nanobubble Phase in Bulk Aqueous Electrolyte Solutions”, Langmuir, 2016, Vol. 32, pp. 11245-11255.
AU: Nanoparticle tracking analysis using Malvern NanoSight LM10 was used to test the effective working of the custom-built MNB generation system. MNB treated DI water had a mean size of 249.8 nm, and SD of 115.8 nm. This data is available elsewhere (Babu, K. S., and Amamcharla, J. K. 2022. Application of micro- and nano-bubbles in spray drying of milk protein concentrates. J. Dairy Sci. In press). However, the data presented is in DI water. Caseins in the GSYs can interfere with the measurement and in the similar size to MNBs, making the quantification of MNBs in the GSY system in terms of size and concentration less realistic. There is no reliable method developed so far to detect and quantify MNBs in milk systems.

Reviewer 2 Report
The Food’s Special Issue, "Emerging Technologies for Improving Properties, Shelf Life and Analysis of Dairy Products", aims to present articles related to applying different types of emerging technologies in the manufacture and preservation of dairy products or dairy components. The manuscript I received for the review describes the application of micro- and nano-bubbles as a tool to improve the rheological and microstructural properties of formulated Greek-style yogurts. It is a novel technology for enhancing the rheological and functional properties of Greek-style yogurt, which aligns with the topic of the Foods Special Issue.
I have some questions and major and minor objections to this work.
L73: Names of microorganisms should be written in italics.
L132: What methods specifically were used to test TS and protein content? What pH meter and electrode was the pH measured with?
L135,140,168,188,197,212,233,243,247,248,250,255,268,309,312,316,317,363,366: please cite the author of the publication. Please review the entire manuscript and check for similar mistakes.
L172: Please describe what methods were used. I would also request an explanation of the number of experiments and the number of trials and replicates.
L177: Were pH, protein content, TS, and titratable acidity tested only in formulated GSY? Why were these parameters not measured during storage? Could the pH value changing during storage affect other parameters, such as syneresis?
L288, 290: Yoghurts or yogurts? Please standardize the writing throughout the manuscript.
L290: Was statistical analysis performed between results from weeks of storage time? Or between types of yogurt samples at different storage times? Does the number n=4 refer to the sample counts in each test group? This would need to be clarified because the results shown in the table are not clear.
L335: It would be more correct to write that "results in table 1 suggest..." rather than "table 1 suggests". Please correct.
L374: It is unfortunate that the authors did not perform a sensory evaluation of the yogurts. The results of the sensory evaluation would have provided additional, very interesting findings for the study.
Author Response
The Food’s Special Issue, "Emerging Technologies for Improving Properties, Shelf Life and Analysis of Dairy Products", aims to present articles related to applying different types of emerging technologies in the manufacture and preservation of dairy products or dairy components. The manuscript I received for the review describes the application of micro- and nano-bubbles as a tool to improve the rheological and microstructural properties of formulated Greek-style yogurts. It is a novel technology for enhancing the rheological and functional properties of Greek-style yogurt, which aligns with the topic of the Foods Special Issue.
I have some questions and major and minor objections to this work.
L73: Names of microorganisms should be written in italics.
AU: This was changed. (Line 74)
L132: What methods specifically were used to test TS and protein content? What pH meter and electrode was the pH measured with?
AU: This was added. (Line 140-142)
L135,140,168,188,197,212,233,243,247,248,250,255,268,309,312,316,317,363,366: please cite the author of the publication. Please review the entire manuscript and check for similar mistakes.
AU: Changed throughout the manuscript, thank you.
L172: Please describe what methods were used. I would also request an explanation of the number of experiments and the number of trials and replicates.
AU: This was added. (Line 131-132)
L177: Were pH, protein content, TS, and titratable acidity tested only in formulated GSY? Why were these parameters not measured during storage? Could the pH value changing during storage affect other parameters, such as syneresis?
AU: Total protein, TS are not expected to change during storage and hence weren’t measured. TA was measured before and after the 4-week storage. This data was updated in the manuscript. (Line 188-190)
L288, 290: Yoghurts or yogurts? Please standardize the writing throughout the manuscript.
AU: This was standardized throughout the manuscript.
L290: Was statistical analysis performed between results from weeks of storage time? Or between types of yogurt samples at different storage times? Does the number n=4 refer to the sample counts in each test group? This would need to be clarified because the results shown in the table are not clear.
AU: A repeated measure (storage) analysis was used for statistical analysis. Control and MNB-treated samples were compared at each storage time. Analysis of variance and least square means at α = 0.05 were used to identify and differentiate means of the significant main effects and interactions; n=4 refers to the total number tested.
L335: It would be more correct to write that "results in table 1 suggest..." rather than "table 1 suggests". Please correct.
AU: This was added.
L374: It is unfortunate that the authors did not perform a sensory evaluation of the yogurts. The results of the sensory evaluation would have provided additional, very interesting findings for the study.
AU: We agree that an in-depth sensory analysis would have been interesting. However, the scope of this research was limited to exploring the rheological and microstructural changes upon MNB incorporation. We do expect to see improved mouthfeel, correlating to the viscosity data.

Reviewer 3 Report
The authors carried out a study entitled "Application of micro- and nano-bubbles as a tool to improve the rheological and microstructural properties of formulated Greek-style yogurts"
In general, the manuscript is well written and brings news, with everything, revisions need to be carried out.
Authors should note that the scientific name of microorganisms, plants or animals must be written in italics, in addition, if a name appears for the first time in the text, it must be written in full.
there are parts of the text that the paravra CO2 o 2 does not exta subscript line 72, eg. Please correct this, this applies to the other compounds.
in some parts of the discussion the authors explore the results more, as is the case of item 3.4, meanwhile, items 3.5 and 3.6 not so much, what is the reason?
The conclusion of a paper should show the main findings and novelties, I think the authors could improve the conclusion, for example I don't see the need to show or report results that were discussed in previous sections.
Author Response

(The authors gave the same response as above.)

Round 2
Reviewer 2 Report
I accept the manuscript in present form.